# Chronic Facial Pain in Fibromyalgia: May ElectroMagnetic Field Represent a Promising New Therapy? A Pilot Randomized-Controlled Study

**DOI:** 10.3390/ijerph20010391

**Published:** 2022-12-26

**Authors:** Marco Storari, Nicoletta Zerman, Benedetta Salis, Enrico Spinas

**Affiliations:** 1Department of Surgical Science, College of Dentistry, University of Cagliari, 09124 Cagliari, Italy; 2Department of Surgery, Dentistry, Pediatrics and Gynaecology, University of Verona, 37124 Verona, Italy

**Keywords:** chronic facial pain, electromagnetic field, fibromyalgia

## Abstract

Fibromyalgic Syndrome is an important public health burden and affects up to 5% of the world population. It requires a complex treatment plan, possibly including antidepressants, anticonvulsants and benzodiazepines, which may in turn affect the patients’ quality of life: hence the need to find additional therapies. The current pilot randomized-controlled study analyzes the effect of electromagnetic field locally administered as add-on therapy in the treatment of cervico-facial pain in patients with fibromyalgic syndrome. 17 patients were selected and low-frequency electromagnetic field was applied via small patches worn in the neck area, between vertebrae C3–C4. Patients were divided into 2 groups, Treated, receiving the therapy, and Placebo, receiving an identical device which was not working,, with respectively 8 and 9 patients. The whole follow up period was 12 months and facial/cervical pain levels were rated using VAS scale. Significant differences were found between patients who received placebo and those treated. Treated patients showed statistically significant improvements in facial/cervical pain at each time-point, both with respect to the previous one and if compared to placebo. In conclusion, low frequency electromagnetic field emerged as beneficial in treating cervico-facial pain in patients with Fibromyalgic syndrome, with no side effects.

## 1. Introduction

Fibromyalgic Syndrome (FS) is a disease that commonly affects the general population. It represents a complex painful syndrome. Long-lasting, widespread and symmetric non-articular musculoskeletal pain are the major features. Furthermore, for diagnosis, generalized tender points at specific anatomical sites are required [1,2]. FS affects up to 5% of the general population [3], it involves mostly women [4] and it can occur at any age [5]. The head and neck region is usually involved. Cervical pain, and even facial pain, especially in terms of masticatory fatigue, are extremely common in FS. Comorbidities are also extremely common, even in the field of orofacial pain, as up to 80% of patients with FM also suffer from headaches or temporomandibular disorders [6]. Fatigue, sleep disturbances, mood disorders and cognitive impairment are also responsible for the low quality of life of affected patients [6,7,8]. Drugs, physical therapy and psychological therapy are the first line therapies in FS [9,10]. Antidepressants, anticonvulsants and opioids are the medications most commonly prescribed [11,12]. These, in turn, tend to significantly impact patients’ quality of life. Conversely, nerve blocks and surgery, by virtue of their invasivity, are recommended only in patients in whom first line therapies have failed [13].

On the other hand, non-invasive brain stimulation techniques have already been shown to be useful in treating chronic pain [14,15]. The aim of non-invasive brain stimulation techniques is to modify the neuronal magnetic fields in order to alter their activity [16]. In this scenario, Electro-Magnetic Fields (EMF), delivering low and very low frequency waves, appeared able to functionally modify endogenous fields within cells and tissues [17,18]. Several authors showed how different targets can be addressed with EMF. In fibroblasts [19] and in chondrocytes [20], respectively, the production of collagen and cartilage glycosaminoglycans was found to be increased after EMF delivery. Similarily, EMF was demostrated to improve osteoblasts’ osteogenic activity [21]. Furthermore, EMF was reported to be able to modulate hormones and neurotransmitters in the interaction with their own targets [22]. Improvements in muscular performance were also observed [23], as well as a better recovery in heart rate and serum lactate levels [24].

In a previous pilot study, we proved the potential effectiveness of low and very low EMF as adjunct treatment in the therapy of chronic orofacial pain in patients with FS and orofacial neuropathic pain [25]. With these premises, a pilot randomized-controlled trial was selected to define more reliably the effectiveness of local EMF in the therapy of cervico-facial pain in patients with FS. Hologies-Orofacial Neuropathic Pain (HO-ONP), [Hologies Srl, Italy, 2159267], was added as adjunct therapy to drugs in patients with FS to manage pain. The properties of HO-ONP have been previously described elsewheres [25].

## 2. Materials and Methods

The purpose of our trial was to analyze the usefulness over time of HO-ONP in facial and cervical pain treatment in individuals with FS as additional therapy to first line medications.

The current study is a pilot randomized and controlled trial. The good clinical practice guidelines for Randomized-controlled trials were followed [26]. Search methods were also implemented with Consort statements [27] in order to test overlap between two relevant search methods. The flowchart has been designed and the recruitment process has been clarified (Figure 1).

Inclusion and exclusion criteria are reported in Table 1. For inclusion, candidates had to satisfy the following conditions: (a) >18 years old, (b) signed informed consent, (c) diagnosis of FS according to the criteria of the American College of Rheumatology [1], (d) complaints of facial and cervical pain due to FS. Additionally, to be included, patients had to show incomplete responsiveness to appropriate medical therapy; this means no more than only a partial recovery from symptoms after 6 months of recommended treatments [10,11]. Exclusion criteria included: (a) psychiatric disorders, (b) TMDs diagnosed according to the Axis I of Research and Diagnostic Criteria/Temporomandibular Disorders (RDC/TMD) [28], (c) chronic orofacial pain of different nature, such as odontalgic pain and neuropathic pain. Moreover, those patients who were addicted to psychotropic substances were also excluded, as were those who had been prescribed non-recommended therapies or who showed high responsiveness rates to recommended medications.

Individuals of both genders were recruited to participate. The initial sample numbered 32 individuals, of whom 29 were women and three men. All the patients were recruited at the Pain Center of the University of Cagliari. All the patients were sent to us with the diagnosis of FS and with complaints of cervical and facial pain. 15 individuals were then excluded because of incompatibility with the inclusion and exclusion criteria, 12 because of concomitant TMDs or neuropathic pain and 3 because of low compliance. Finally, a total of 17 individuals were selected, all females, between 30 and 78 years old (Figure 2).

Patients were randomized by the independent coordinator (ES) who was not involved in the selection, treatment or follow-up of patients. A list of patients was created and a computer-generated sequence randomized them in a 1:1 ratio to either medical therapy or placebo. Randomization codes were obtained after patients were recruited to the trial. The randomization sequence was held by the independent coordinator (ES). Both participants and researchers (MS and BS) were unable to foresee the assignment. All participants’ data was pseudoanonymised with personal information removed and replaced by a coded identifier. Because of the nature of the study, both investigators (MS and BS) and patients were blinded regarding their allocation. Individuals were firstly grouped in subcategories according to age, such as 18–40; 41–60; +61, in order to have in both groups at least 1 patient from each of these subcategories, in equal number as far as possible.

(a) Treated Group: 8 patients received EMF-delivering patches; one patch for each patient was delivered; (b) Control Group: 9 patients received placebo patches, absolutely indistinguishable from functioning ones without being informed.

In order to avoid a statistically significant difference between the 2 groups at the beginning of the trial, pain levels were evaluated by the program Ranksum, described below, and drug intakes were listed, in order to have similar drugs and dosages in both groups.

The current trial was developed according to the good clinical practice guidelines of the Declaration of Helsinki [28]. The study was approved by the Medical Direction Board of the Surgical Sciences Department of the University of Cagliari [PROT.PG/2020/13742].

Two researchers (MS and BS) followed the patients during the trial, that is for a year, at the Orofacial pain center of the University of Cagliari. The whole trial was developed under the supervision of an independent coordinator (ES). To define agreement rates between the researchers, the kappa test was applied and it emerged as moderate (0.55). Any disagreements were discussed by all the authors.

The low and very low frequency EMF, kept at very low intensity to maintain the localized effect, was delivered through a system of nanocrystals interacting with semiconductors targeted with appropriate signals. The signal traced the calcium ion’s own signal to create a locally selected and controlled incoherence with the endogenous calcium itself. The objective was to reduce the calcium flow by intervening indirectly on the pro-nociceptive modifications induced in both voltage-gated and ligand-gated calcium channels. As we exploit wavelengths built to mirror calcium’s, a very important ion for several functions, we use very low-intensity EMF so as not to disturb other calcium-related activities. This technique involves only the selected ionic activities that express pain signals downstream of the nerve branch on which the EMF was induced.

The trial was structured through six consecutive steps. (1) Medical examination: patients’ signs and symptoms were investigated. Pain characteristics were detected and the average pain level was rated on the VAS scale [29]. The VAS scale represented a 100 mm line where individuals had to trace the level of pain [30], approximately, from “no pain” to “the worst ever pain”. Hyperalgesia and allodynia were also evaluated; patients were educated about features and possible side effects of HO-ONP. Lastly patients were asked to give and sign the informed consent. Patients continued to be under the supervision of the same doctor at the Pain Center for the whole period and weren’t allowed to change their initial pharmacological protocol: Duloxetine 30 mg/daily in the morning, Eperisone Hydrochloride 100 mg/once or twice per day, Clonazepam 5/7 drops before bedtime and Magnesium 3 to 4.5 mg/kg per day. (2) Application: EMF was delivered using a small, flexible and comfortably wearable patch which was 2 × 3 cm in size (Figure 3). The patch was applied between vertebrae C3–C4 directly by the researchers (MS and BS). Patients were instructed to sign on the VAS scale the level of pain before wearing the patch, thus at T0 (0 s). Patients were taught about the care of the device and were also advised to change the position of the patch once per week to avoid any possibility of tolerance. (3) Discharge: during the first day after the application patients were instructed to sign the pain level on the VAS scale after 1 h, then after 6 h and lastly before sleeping, and to pay careful attention to any side effect. Subsequently, and every day for the next 12 months, patients had to indicate their pain level after waking, during the afternoon and before sleeping. (4) 1st check (T1): patients were recalled to the department one month after the application. During the examination any improvements in symptomatology and side effects were noted. After that moment, and for the following year, patients were examined once a month. Data about pain levels, general symptoms and possible side effects were collected during each visit. (5) 2nd check (T2): There was an intermediate check up 6 months after the application, mostly for statistical purposes. (6) Last check (T4): patients were recalled for the last time 12 months after the application. Overall benefits and side effects of the therapy were evaluated and registered.

Definitely the 17 patients were deemed satisfactory to allow for an adequate value of the non-parametric statistical analysis selected for the study as the aforementioned was a pilot study.

Data were organized into a spreadsheet (MS Excel Office 365 MSO). The significance of changes in VAS score were evaluated using the Ranksum Statistical Method in particular, for independent samples, the test of the sum of the ranks of Wilcoxon or Mann-Whitney, while, for dependent samples, the Wilcoxon test by signs [31,32]. Statistical significance was set at *p* < 0.05 and the software Stata (release 16, College Station, TX: StataCorp LLC) was used to analyze all data.

## 3. Results

The present trial evaluated the responsiveness rate of a total of 17 individuals with FS to a new nanotechnological device based on Low-Frequency Electromagnetic Waves, called “HO-ONP”.

Patients belonging to the Treated Group mostly reported improvements in pain intensity immediately after the application and confirmed the same at all the subsequent checks, appreciable by observing signed VAS scales. However, symptoms in the other parts of the body did not improve. By contrast, the Control Group patients did not report significant improvements. In any case, all individuals, regardless of the group they belonged to, reported any complaints concerning the ergonomics of the patch. No side effects were reported by any patients.

Non-parametric tests were carried out in order to verify whether the investigated treatment was effective. This test was chosen due to the small sample size. Indeed this particular type of test, based on ranks, does not need to make assumptions about the distribution of the population. To understand whether there are differences between the control group (Placebo) and the one that received the treatment (Treated), the test we carried out, for independent samples, was the sum of the ranks of Wilcoxon or Mann-Whitney. It was thus possible to verify whether there were statistically significant differences between the two groups before and during the different stages of administration of the treatment. The results summarized in Table 2 indicate that before administration (T0) there were no statistically significant differences between the two groups (*p* value 0.6644). Following the administration the data show that there was a statistically significant difference (*p* value 0.0384) between the two groups already in the first month of treatment (T1). The difference remained significant (*p* value 0.0433) even after six months (T2), and still persisted at a distance of one year (T3) (*p* value 0.0269).

Following this first verification, which informed us that after treatment the treated and control groups differed, we proceeded with further analyses based always on non- parametric tests, but in this case for dependent samples. We investigated whether the perception of pain in the different stages of treatment had changed within the groups, using the Wilcoxon test by signs (Table 3). In the Placebo group, the perception of pain over time was never significantly different from the T0 (*p* value 0374, 0.5529, 0.9528, respectively at T1, T2 and T3). By contrast, among patients who received the treatment, statistically significant differences emerged from the first month of administration and persisted over the follow up period (*p* value 0.0117, 0.0357, 0.0357, respectively at T1, T2 and T3).

A simpler analysis can be seen in Figure 4. Through the analysis of the above reported statistical data, no relevant differences emerged in the Placebo Group while a significant decrease in pain levels appeared in the Treated Group. Due to the asymmetric distribution of pain levels among patients in each group, the main value is represented by the P50, the median. According to this measure, the median pain levels in the Treated group fell progressively from 6.7 at T0 to 4.8 at T1, 3.0 at T2 and 1.5 at T3. A similar pattern also emerged when P25 and P75, as well as the minimum, were taken into consideration.

## 4. Discussion

The authors’ purpose was to establish the validity of the HO-ONP as adjuvant therapy in the treatment of pain in the orofacial district in patients affected by FS.

EMF has proved to be a valid, noninvasive and accessible method used for the treatment of pain, and its effectiveness in the management of various painful and inflammatory disorders has been well assessed [33,34,35,36,37]. Sutbeyaz et al. [38] showed that Low-frequency EMFT is able to improve mobility and decrease both pain and fatigue in patients with fibromyalgia, and these outcomes were further confirmed by Sampson et al. [39]. Focusing on the facial and cervical regions, we demonstrated in a first pilot study the effectiveness of HO-ONP patch in the treatment of FS and Orofacial Neuropathic Pain conditions [25]. In particular, patients with FS showed major improvements over time. In the present study we conducted a randomized-controlled trial focused on FS to find out whether HO-ONP is reliable and efficacious as adjuvant therapy compared to placebo. We found a significant difference between the treated and the control patients, already a week after the application, with persistent benefits reported during the whole period of the observation and after each check. Indeed, the median shows important improvements in treated patients, from a value of 6.7 to 1.5 after one year of therapy, while no significant changes appeared in untreated patients. Several other parameters showed a similar trend, with a net decrease in the Treated group and no significant changes in the Placebo group. Only data referring to the Maximum value did not change, both looking at each group and between the two groups, meaning that the therapy is not effective in all cases. Comparing the two groups, statistically significant differences emerged in pain levels over time, already after one month of treatment, and then after six months and one year. A further confirmation was given by analysis of pain perception over time in each single group. Indeed, only the Treated group showed a significant decrease of pain, while no differences appeared in the Placebo group.

Interestingly, but as might have been expected, these effects were felt only in treated parts, that is above the shoulders, while the overall symptomatology remained unchanged in other body regions.

Despite these important, but localized, improvements, patients who received the functioning device needed to continue their current drug-based therapy to reduce complaints related to the rest of the body. This aspect however requires additional analysis because patients belonging to the Treated Group decreased the physical therapy exercises usually done to reduce cervical pain and improve neck and shoulder movements. It may be reasonable to assume that applying the same patches to multiple body parts could bring benefits in symptoms while reducing drug dosages. No individual who took part in the study complained about the ergonomics of the device. The patch appeared practical, it did not [seem to] inflict any discomfort and it does not require any sort of power supply, unlike other medical equipment delivering electromagnetic waves. Last, but not least, no side effects were reported by patients in either group.

An important feature of our study concerns the population studied. Indeed, only female individuals met the inclusion criteria and were recruited into the analysis. This feature is justified by the higher attested prevalence of FS in women than in men. In a recent update of the literature review on the prevalence of FS, it was found that this pathology has a frequency between 2.4% and 6.8% in women, especially after menopause, compared with a prevalence between 0, 2% and 4.7% of the general population [40]. Similar results were confirmed by Queiroz et al. who demonstrated that there is a 3:1 ratio in favor of women in the incidence of FS [41].

Furthermore, despite the small number of the sample due to the application of selective inclusion and exclusion criteria in the face of an uncommon pathology, we consider the results achieved to be significant. Statistical analysis has in fact highlighted that adjuvant therapy with EMF can be inserted as a valid therapeutic option additional to the more traditional pharmacological therapy in cranio-cervical-facial pain in patients with FS.

## 5. Conclusions

In conclusion we can support the benefits given by the Holographic Information Transfer Analgesic Patch in alleviating head and neck pain in patients affected by FS. Additionally the localized effect requires further research to investigate possible systemic coverage in order to improve the overall lifestyle of such patients.

## Figures and Tables

**Figure 1 ijerph-20-00391-f001:**
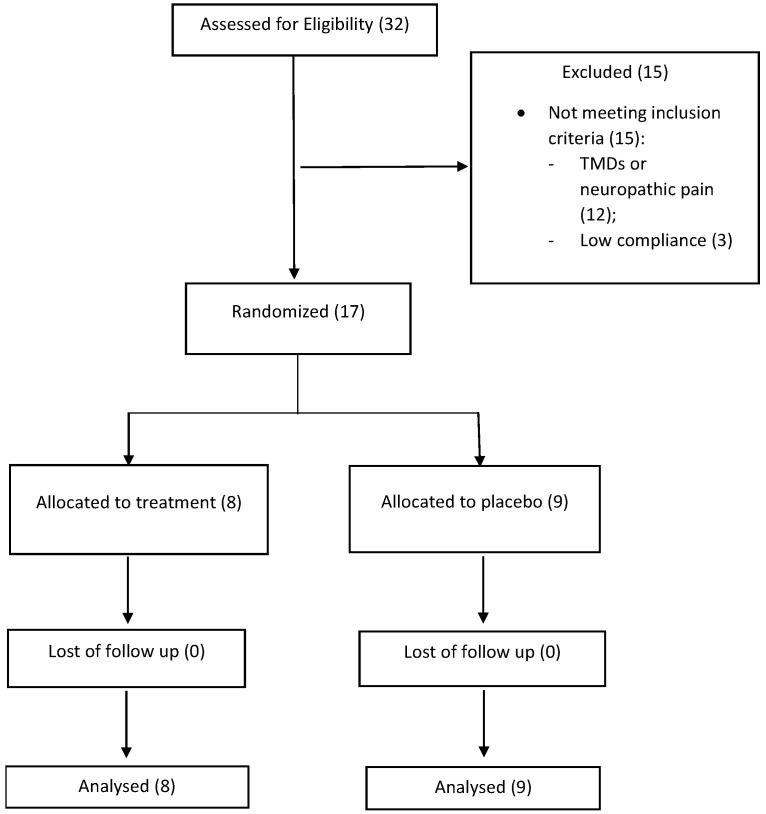
CONSORT. The table analyzed the methodological process of patients recruitment for the randomized controlled trial.

**Figure 2 ijerph-20-00391-f002:**
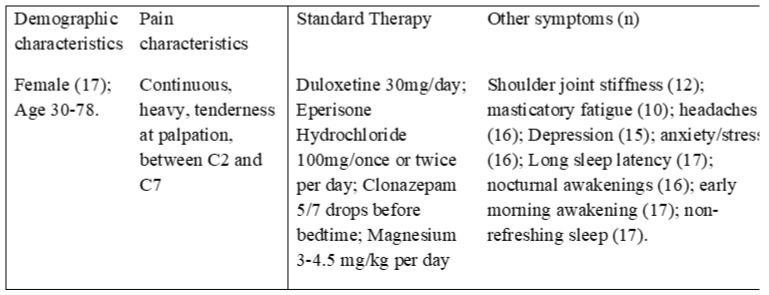
General characteristics of the included individuals.

**Figure 3 ijerph-20-00391-f003:**
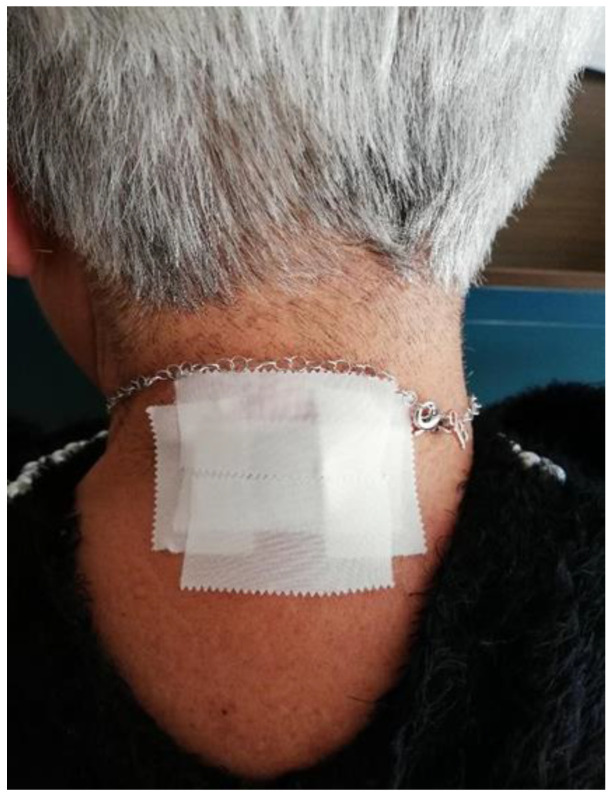
Standardized Model of Application of HO-ONP patch.

**Figure 4 ijerph-20-00391-f004:**
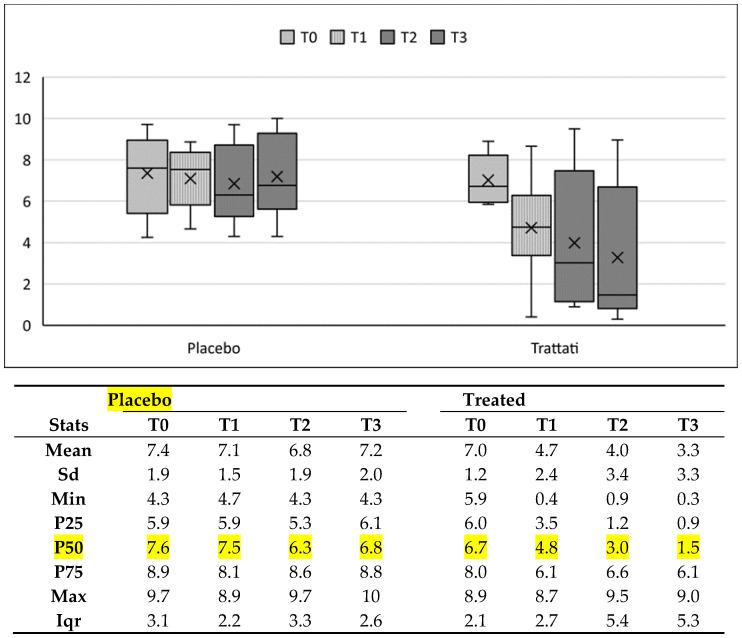
VAS scale Box, Placebo group and treated, from T0 to T3. The boxplot shows the analysis of the below reported statistical data; the main value is represented by the P50 due to the asymmetric distribution of pain levels among patients in each group. It is easily seen that there are no significant differences in the Placebo Group while a significant decrease appears in the Treated Group, from 6.7 at T0 to 4.8 at T1, 3.0 at T2 and 1.5 at T3.

**Table 1 ijerph-20-00391-t001:** Inclusion and exclusion criteria.

Inclusion Criteria	Exclusion Criteria
>18 years old; Both genders signed informed consent; FS diagnosed according to the American College of Rheumatology’s guidelines; facial and cervical pain due to FS under appropriate pharmacological treatment albeit without complete responsiveness.	Psychiatric disorders which influence compliance with the therapy TMDs diagnosed according to the Axis I of Research and Diagnostic Criteria/Temporomandibular Disorders (RDC/TMD); Other chronic orofacial pains; Addiction to psychotropic substances; Without recommended pharmacological treatment; With recommended pharmacological treatment and with high responsiveness rates to medications.

**Table 2 ijerph-20-00391-t002:** Test results, non-parametric for independent samples, of the sum of the ranks of Wilcoxon and Mann-Whitney. Comparison between treated and placebo groups over time. No significant statistical difference between Treated and Placebo at T0 (*p* > 0.6644). Statical significance appeared at both T1 (*p* = 0.0384) and T2 (*p* = 0.0433), and greatly increase at T3 (*p* = 0.0269).

Two-Sample Wilcoxon Rank-Sum (Mann-Whitney) Test				
**H** **0:**	T0(Placebo) = T0(Treated)	T1(Placebo) = T1(Treated)	T2(Placebo) = T2(Treated)	T3(Placebo) = T3(Treated)
**Z =**	0.434	2.070	2.021	2.213
**Prob >** **|z| =**	0.6644	0.0384	0.0433	0.0269

**Table 3 ijerph-20-00391-t003:** Test results, non-parametric for dependent samples, signs of Wilcoxon ranks. Comparison of the responsiveness rate within each group over time in relation to T0.

Wilcoxon Signed-Rank Test		
	Placebo	Treated
**H0:**	T0 = T1	T0 = T1
**Z =**	0.889	2.521
**Prob > |z| =**	0.3743	0.0117
**H0:**	T0 = T2	T0 = T2
**Z =**	0.593	2.100
**Prob > |z| =**	0.5529	0.0357
**H0:**	T0 = T3	T0 = T3
**Z =**	0.059	2.100
**Prob > |z| =**	0.9528	0.0357

## Data Availability

Data sharing not applicable due to privacy.

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
