# Peer review of "Chronic Facial Pain in Fibromyalgia: May ElectroMagnetic Field Represent a Promising New Therapy? A Pilot Randomized-Controlled Study"

_ijerph, 2022, doi:10.3390/ijerph20010391_

Round 1
Reviewer 1 Report (Previous Reviewer 1)
The authors resubmitted a pilot RCT on using EM field for treatment of chronic facial pain in patients with fibromyalgia.
I have the following comments:
1. There is insufficient description of the important parts of a RCT, e.g. outcome measures, sample size calculation etc. It simply did not follow the CONSORT statement, which is the gold standard of a RCT currently. A "pilot" study should not be missing these as well as these determines the whole study design.
The authors did not address my previous concerns. I would not recommend acceptance.
Author Response
Reviewer 1 - Round 1 (28 august 2022)
Dear reviewer,
thank you for the report and the suggestions, we found them extremely helpful to improve our manuscript. We apologize to not be able to satisfy your request about CONSORT statement as we followed different guidelines. We considered the Good Clinical Practice (GCP) in Randomized Controlled Trial (RCT) the best option for the current trial. We know the CONSORT scheme and we used it previously in other trials, but we chose the Good Clinical Practice (GCP) in Randomized Controlled Trial (RCT) because of the paucity of data and patients.
Here the answers to your notes:
- The authors presented the pilot data of a randomized clinical trial to investigate if EM field may help in chronic facial pain in fibromyalgia.
I have the following comments:
- Even though it is a pilot study of a RCT, I would have expected the study design and reporting followings the CONSORT statement, while just presenting the initial findings. However, it did not follow the standard of RCT.
We did not follow the CONSORT statement as we followed the Good Clinical Practice (GCP) in Randomized Controlled Trial (RCT). The informed consent was delivered to each patients; Accurate inclusion and exclusion criteria were selected; The approval of the ethical committee was obtained; the randomization was performed anonymously; a comparable number of patients was collected in both group and no differences between patients in the 2 groups were observed; patients were followed over time according the good clinical practice guidelines.
- What are the outcome (primary and secondary) measures? Not reported
The outcome of our trial was explicated among lines 72-74 (“The purpose of our trial was to analyze the usefulness over the time of HO-ONP in facial and cervical pain treatment in individuals affected from with FS as additional therapy to first line medications”).
- Sample size calculation is most crucial. It was not described.
We do not have an established number of individual to start the trial. We defined firstly inclusion/exclusion criteria and then we applied such criteria to the whole cohort of patients sent to us by the Specialist doctors of the Pain Center.
Round 2 (October 2022)
The authors did not address my concern of the previous review. I could not recommend acceptance.
We are not able to further satisfy the request as we have previously motivated our choice. We did not refer to the CONSORT guidelines but to the Good Clinical Practice (GCP) in Randomized Controlled Trial (RCT) [Abu Talha K. Good Clinical Practice (GCP) in Randomized Controlled Trial (RCT). April 2016. Conference: FMHS Research Seminar 2016]. We considered the Good Clinical Practice (GCP) in Randomized Controlled Trial (RCT) the best option for the current trial. We know the CONSORT scheme and we used it previously in other trials, but we chose the Good Clinical Practice (GCP) in Randomized Controlled Trial (RCT) because of the paucity of data and patients.
Round 3 December 2022)
The authors resubmitted a pilot RCT on using EM field for treatment of chronic facial pain in patients with fibromyalgia.
I have the following comments:
- There is insufficient description of the important parts of a RCT, e.g. outcome measures, sample size calculation etc. It simply did not follow the CONSORT statement, which is the gold standard of a RCT currently. A "pilot" study should not be missing these as well as these determines the whole study design.
The authors did not address my previous concerns. I would not recommend acceptance.
To address the reviewer’s requests we definitely analyzed and inserted into the manuscript the CONSORT’s guidelines. The comparation between CONSORT and the Good Clinical Practice (GCP) in Randomized Controlled Trial (RCT) highlited a clear overlap between the 2 methods. We provided to add the CONSORT table. The methods of selection and exclusion of the sample are described between lines 104 to 146.
Reviewer 2 Report (Previous Reviewer 2)
I take note of the explanations. I think that it would be worth checking the symptoms with which the patient reported, despite the fact that FS has such variable and non-specific symptoms.
VAS is a visual and non-drawable scale. Pain drawing enables the patient to mark areas of pain in the head, neck and whole body diagrams.
Author Response
Round 1 and 2 (August and October 2022)
Dear reviewer,
Thank you for the report and the suggestions, we found them extremely helpful to improve our manuscript. We provided the corrections suggested for every part of the study, as you can find in the revised manuscript. The English level was overall ameliorated. You can find the same comments besides the manuscript with the appropriate references in the text.
- Authors may want to clarify the location and nature of cervicofacial pain, standard therapy, and comorbid symptoms which each participant had at the beginning of the study. It can be summarized in one table because the number of participants is only 17.
The purpose of the trial was to investigate the effectiveness of EMF on Cervico-facial pain. We did not evaluate the effect of EMF on any other symptoms, because it was not in our aims. Pain characteristics and the standard therapy were described in materials and methods (table 1 and lines 142-144). The below table was built to summarize demographic data of included individuals and was inserted in Materials and methods.
|
Demographic characteristics |
Pain characteristics |
Standard Therapy |
Other symptoms (n) |
|
Female (17); Age 30-78. |
Continuous, heavy, tenderness at palpation, between C2 and C7 |
Duloxetine 30mg/day; Eperisone Hydrochloride 100mg/once or twice per day; Clonazepam 5/7 drops before bedtime; Magnesium 3-4.5 mg/kg per day |
Shoulder joint stiffness (12); masticatory fatigue (10); headaches (16); Depression (15); anxiety/stress (16); Long sleep latency (17); nocturnal awakenings (16); early morning awakening (17); non-refreshing sleep (17). |
- Authors should clarify methods for evaluating sleeping, headache, functional limitation in movements, emotion and so on.
All the methods to evaluate pain are based on subjective reports. VAS scale is one of the best possible. Thus, we limited the research to this one, also because pain in such patients is diffuse to all head and neck district and difficult to isolate in the drawing method. All the other symptoms out of pain have been evaluated only during the interview because it was not in the aim of the study to evaluate other symptoms than pain. Headaches have been analyzed asking frequency (times per month) and intensity (VAS scale) while functional limitations, sleep disorders and mood disorders were reported based on patients’ self report. These symptoms we no further investigated with objective methods.
- Table 1 (inclusion and exclusion criteria): It's hard to understand because the last line is merged.
As suggested, to make more comprehensive the inclusion/exclusion criteria table, I moved the whole table in a single space.
- “Authors should add table or graph, which shows the results for sleeping, headache, functional limitation in movements, emotion and so on” & “Although authors demonstrate that EMF improves sleep, function, and mood in addition to cervicofacial pain, no evidence was shown in the manuscript. Authors should discuss based on the data and its analysis.”
Since we didn’t use any objective method of evaluation over time for Sleep disorders, headaches, functional limitations and mood disorders, our analysis may be only based on the simple report of improvements of such symptoms.
- Authors may want to suggest their thoughts regarding the mechanism how EMF improves cervicofacial pain. Also, they may suggest why it did not affect pain below the neck. Do authors consider that EMF is a beneficial treatment even if it does not reduce pain below the neck?
The purpose of the trial was to evaluate the potential benefits of the local administration of EMF. We focused only to induce a localized administration of EMF also because this trial was performed in the dental/maxillofacial sector. At the same time, as we exploit wavelengths built mirror-like to calcium, a very important ion for several functions, we use very low-intensity EMF to not disturb other calcium-related activities. This technique involves only the selected ionic activities that express pain signals downstream of the nerve branch on which the EMF was induced. The EMF works at the same time via the oxyhemoglobin signal which allows in turn a more prompt and powerful effect in the administered area. In this sense is explained the localized effect of our therapy. Based on such premises, we consider potentially beneficial EMF because it positively answered our research question. [Seto YJ, Hsieh ST. Electromagnetic induced kinetic effects on charged substrates in localized enzyme systems. Biotechnol Bioeng. 1976 Jun;18(6):813-37.]
- Since two factors, treatment and time, are delt with, a two-way statistical test should be used, not one-way method like Mann-Whitney.
The statistical methods used are, for independent samples, the test of the sum of the ranks of Wilcoxon, also called Mann-Whitney test, while, for dependent samples, the Wilcoxon test by signs. Such tests were chosen because of the very low number of the sample. At the same time, in the trial the variable time was considered as a discrete and not continuous variable, namely 1 month, 3 months, 6 months and 1 year. By contrast, pain was analyzed with a continuous cumulative distribution. The Mann-Whitney test is a non-paracontinuous-level test, in contrast to the t-tests and the F-test; it does not compare mean scores but median scores of two samples. Thus it is much more robust against outliers and heavy tail distributions. The Mann-Whitney test compare, in our case, the difference between 2 independent variables independently at each time-point. To follow over time the effect of the therapy we chose the Wilcoxon test by signs in which, again, time is a discrete variable.
(Fay MP, Proschan MA. Wilcoxon-Mann-Whitney or t-test? On assumptions for hypothesis tests and multiple interpretations of decision rules. Stat Surv. 2010 ; 4: 1–39)
- English language:
All the mistakes have been corrected and the manuscript underwent an extensive revision of the English.
- Abbreviations
Abbreviations have been modified as suggested.
Round 3 (December 2022)
I take note of the explanations. I think that it would be worth checking the symptoms with which the patient reported, despite the fact that FS has such variable and non-specific symptoms.
VAS is a visual and non-drawable scale. Pain drawing enables the patient to mark areas of pain in the head, neck and whole body diagrams.
We thank the reviewer for the comments and the revision to the manuscript. Our aim was to analyzed the effect of the EMF as add-on-therapy exclusively referred to cranio-facial district in patients affected from FS. Potential benefits in other districts were out of the aims of the work.
All the methods to evaluate pain are based on subjective reports. VAS scale is one of the best possible. Thus, we limited the research to this one, also because pain in such patients is diffuse to all head and neck district and difficult to isolate in the drawing method. All the other symptoms out of pain have been evaluated only during the interview because it was not in the aim of the study to evaluate other symptoms than pain.
Reviewer 3 Report (New Reviewer)
Dear Author
I congratulate you on the development of clinical work on a very frequent health problem for the population.
I recommend you to improve the version of the article:
- Despite being a pilot study, declare the calculation of the chosen sample and the statistical power achieved.
- Develop in the discussion the prevalence of the disease that affects women more than men to justify the disparity of the sample between women and men.
Best regards
Author Response
Dear Author
I congratulate you on the development of clinical work on a very frequent health problem for the population.
I recommend you to improve the version of the article:
- Despite being a pilot study, declare the calculation of the chosen sample and the statistical power achieved.
- Develop in the discussion the prevalence of the disease that affects women more than men to justify the disparity of the sample between women and men.
Dear reviewer,
Thank you for the report and the suggestions, we found them extremely helpful to improve our manuscript. We provided the corrections suggested for every part of the study, as you can find in the revised manuscript.
17 patients were deemed satisfactory to allow for an adequate value of the non-parametric statistical analysis selected for the study as the aforementioned was a pilot study. In the discussion, it was specified why only female patients were recruited, in accordance with the recent bibliography on the subject. Furthermore, the value of the obtained results, including statistical ones, was amply justified, underlining how this therapy can be inserted as a valid add-on therapy to traditional drugs and allowing a significant local reduction of chronic neck and facial pain and of the physical therapy exercises usually done to reduce cervical pain and improve neck and shoulder movements.
Round 2
Reviewer 1 Report (Previous Reviewer 1)
The authors could not amend to an expected standard of publishing a RCT.
In my opinion, this is below an acceptable level for most journals
This manuscript is a resubmission of an earlier submission. The following is a list of the peer review reports and author responses from that submission.
Round 1
Reviewer 1 Report
The authors presented the pilot data of a randomized clinical trial to investigate if EM field may help in chronic facial pain in fibromyalgia.
I have the following comments:
1. Even though it is a pilot study of a RCT, I would have expected the study design and reporting followings the CONSORT statement, while just presenting the initial findings. However, it did not follow the standard of RCT.
2. What are the outcome (primary and secondary) measures? Not reported
3. Sample size calculation is most crucial. It was not described.
The flaws in the methodology made the interpretation of the result questionable.
Author Response
Dear reviewer,
thank you for the report and the suggestions, we found them extremely helpful to improve our manuscript. The English level was overall ameliorated. Here the answers to your notes:
- We do not have an established number of individual to start the trial. We defined firstly inclusion/exclusion criteria and then we applied such criteria to the whole cohort of patients sent to us by the doctors of the Pain Center.
- We did not follow the CONSORT statement as we followed the Good Clinical Practice (GCP) in Randomized Controlled Trial (RCT)
Reviewer 2 Report
A very practical article on fibromyalgia that is difficult to treat. Made according to the randomized research protocol (GCP). From the reviewer's position, I have comments.
I do not list the symptoms of ES after the intervention (there is only a statement that there was no improvement in symptoms in other parts of the body).
Please list the symptoms that were tested before and after in the face, head and neck, because in the results were described the reduction of other symptoms (shoulder joint stiffness, headaches, mastication, etc.), i.e. it must be stated that these symptoms were assessed before the intervention and then after and when.
Especially that the study was based only on a clinical trial and a subjective assessment. Was “Pain drawing” used? The statistical analysis concerned only the VAS in facial and cervical region, which should be clearly emphasized in point 1 of the description of the study method.
Did the authors find side effects? If so, how did they handle it? I am also interested in how the Authors assessed the addiction of patients to psychotropic substances?
The work has a mistake in the title - 'a' is written with a lowercase letter, there are a few spelling mistakes. Therefore, I ask the Authors to check the text before printing, but besides, congratulations and we look forward to working on a larger group so that clinicians can use this method.
The work is ready for publication after taking into account the comments of the reviewer.
Author Response
Dear reviewer,
thank you for the report and the suggestions, we found them extremely helpful to improve our manuscript. We provided the corrections suggested for every part of the study, as you can find in the revised manuscript. The English level was overall ameliorated. You can find the same comments besides the manuscript with the appropriate references in the text.
- All the methods to evaluate pain are based on subjective reports. VAS scale, as drawing, is one of the best possible Thus we limited the research to this one, also because pain in such patients is spreaded at all head and neck district and difficult to isolate. In our opinion it would have made more tiring the experience for patients.
- We took notes of side effects at each appointment.
- The purpose of our trial was to evaluate potential benefits of EMF on pain. We focused on such outcome and we did not consider other symptoms. However, and we thought it was worthy to be mentioned, patients themselves reported us improvements in several other symptoms we firstly did not consider.
- The purpose of our trial was to evaluate potential benefits of EMF on pain. Thus we limited our investigation on benefits and side effects of the device. It was not in the aims of the study to evaluate benefits and side effects of other therapies.
Reviewer 3 Report
The authors investigated the effect of electromagnetic field therapy on cervicofacial pain in patients with fibromyalgia. They demonstrate that the therapy, which was added to the standard therapy, significantly reduced the VAS score as well as other comorbidities including sleep disorders. I suggest that the authors should consider the following points.
Materials and Methods:
Authors may want to clarify the location and nature of cervicofacial pain, standard therapy, and comorbid symptoms which each participant had at the beginning of the study. It can be summarized in one table because the number of participants is only 17.
Since two factors, treatment and time, are delt with, a two-way statistical test should be used, not one-way method like Mann-Whitney.
Authors should clarify methods for evaluating sleeping, headache, functional limitation in movements, emotion and so on.
Table 1 (inclusion and exclusion criteria): It's hard to understand because the last line is merged.
Results:
Authors should add table or graph, which shows the results for sleeping, headache, functional limitation in movements, emotion and so on.
Figure 2: “Tratatti” should be changed to English.
Discussion:
Although authors demonstrate that EMF improves sleep, function, and mood in addition to cervicofacial pain, no evidence was shown in the manuscript. Authors should discuss based on the data and its analysis.
Authors may want to suggest their thoughts regarding the mechanism how EMF improves cervicofacial pain. Also, they may suggest why it did not affect pain below the neck. Do authors consider that EMF is a beneficial treatment even if it does not reduce pain below the neck?
English language:
Authors should have the full text be checked carefully by English native speaker. The manuscript is full of unnatural expressions. Below are just a few examples.
# The authors’ purpose was to established the validity→ to establish
# various distinctive benefitsthat → benefits that
# did not “or only partially” considered → did not consider or only partially did
# Who took a part in the study → took part in
# The patch appeared practical, it do not indicate → it does not
Style:
There are many types in the manuscript. Read your manuscript carefully and correct them. Below are just a few examples.
# electromagnetic field waves ((regarding the emotional → (regarding the
# fi-bromyalgia → fibromyalgia
# The low and very lo → very low
# Capital letters are used in unnatural places. For example, “Figure 1 Standardized model Of…”
Abbreviation:
Define abbreviations on first occurrence and use them consistently. Below are just a few examples.
# Fibromyalgic syndrome (FS): Fibromyalgia syndrome (FMS) seems to be more standard
# FM: Define on first occurrence (Fibromyalgia?), and use it in place of “fibromyalgia”
# EMF: Define on first occurrence (electromagnetic field?)
# MS
# BS
Redundant repetition
Below are just a few examples.
Do not redundantly repeat the figures in the Table. For example, “significant difference (P Value 0.0384)…..”
Do not redundantly repeat the result in the Discussion. For example,”,with a P value, respectively of 0.0384 at T1…”
Author Response
Dear reviewer,
Thank you for the appreciation and the advices. We found extremely helpful your comments. We provided the suggested corrections, as you can find in the revised manuscript. The English level was overall ameliorated. You can find the same comments besides the manuscript with the appropriate references in the text.
- As we said in materials and methods, our purpose was to evaluate the potential benefits of the local administration of EMF. At the same time, as we use wavelenghts built mirror-like to calcium, a very important ion for several fnctions, we use very low intensity EMF. This is why the effect was only localized.
- The purpose of our trial was to evaluate potential benefits of EMF on pain. We focused on such outcome and we did not consider other symptoms. However, and we thought it was worthy to be mentioned, patients themselves reported us improvements in several other symptoms we firstly did not consider.
- We appreciate your comment as we discuss a lot about the statistical method to use. Lastly we decided to these two because of the very low number of the sample and thus to simplify the results. In the next trials of courese we will use other more complex methods.
- All the methods to evaluate pain are based on subjective reports. VAS scale, as drawing, is one of the best possible Thus we limited the research to this one, also because pain in such patients is spreaded at all head and neck district and difficult to isolate. In our opinion it would have made more tiring the experience for patients.
- As suggested, to make more comprehensive the inclusione/exclusion criteria table, I moved the whole table in a single space.
Round 2
Reviewer 1 Report
The authors did not address my concern of the previous review. I could not recommend acceptance
Reviewer 3 Report
The author did not incorporate most of the suggestions in Report 1 in the revised version.
Materials and Methods:
Authors did not reflect the following suggestion in the revised manuscript.
(Suggestion in Report 1) Authors may want to clarify the location and nature of cervicofacial pain, standard therapy, and comorbid symptoms which each participant had at the beginning of the study. It can be summarized in one table because the number of participants is only 17.
Authors did not reflect the following suggestion in the revised manuscript. Mann-Whitney test seems inappropriate to deal with two factors (treatment and time).
(Suggestion in Report 1) Since two factors, treatment and time, are delt with, a two-way statistical test should be used, not one-way method like Mann-Whitney.
Authors did not reflect the following suggestion in the revised manuscript, It seems inappropriate to refer to the improvement in these symptoms without showing methods/data.
(Suggestion in Report 1) Authors should clarify methods for evaluating sleeping, headache, functional limitation in movements, emotion and so on.
Results:
Authors did not reflect the following suggestion in the revised manuscript, It seems inappropriate to refer to the improvement in these symptoms without showing relevant data.
(Suggestion in Report 1) Authors should add table or graph, which shows the results for sleeping, headache, functional limitation in movements, emotion and so on.
Authors did not reflect the suggestion in the revised manuscript.
(Suggestion in Report 1) Figure 2: “Tratatti” should be changed to English.
Discussion:
Authors did not reflect the following suggestion in the revised manuscript. It seems inappropriate to refer to the improvement in these symptoms without showing relevant methods/data.
(Suggestion in Report 1) Although authors demonstrate that EMF improves sleep, function, and mood in addition to cervicofacial pain, no evidence was shown in the manuscript. Authors should discuss based on the data and its analysis.
Authors did not reflect the following suggestion in the revised manuscript.
(Suggestion in Report 1) Authors may want to suggest their thoughts regarding the mechanism how EMF improves cervicofacial pain. Also, they may suggest why it did not affect pain below the neck. Do authors consider that EMF is a beneficial treatment even if it does not reduce pain below the neck?